# *Acinetobacter baumannii*, a Multidrug-Resistant Opportunistic Pathogen in New Habitats: A Systematic Review

**DOI:** 10.3390/microorganisms12040644

**Published:** 2024-03-23

**Authors:** Omar E. Ahuatzin-Flores, Eduardo Torres, Edith Chávez-Bravo

**Affiliations:** 1Posgrado en Ciencias Ambientales, Instituto de Ciencias, Benemérita Universidad Autónoma de Puebla, Edificio IC 6. Ciudad Universitaria, Puebla 72570, Mexico; omare.ahuatzinflores@viep.com.mx; 2Centro de Química, Instituto de Ciencias, Benemérita Universidad Autónoma de Puebla, Puebla 72570, Mexico; eduardo.torres@correo.buap.mx; 3Centro de Investigaciones en Ciencias Microbiológicas, Instituto de Ciencias, Benemérita Universidad Autónoma de Puebla, Puebla 72570, Mexico

**Keywords:** *Acinetobacter baumannii*, multi-resistance, natural habitats, reservoirs

## Abstract

In recent years, humanity has begun to face a growing challenge posed by a rise in the prevalence of antibiotic-resistant bacteria. This has resulted in an alarming surge in fatalities and the emergence of increasingly hard-to-manage diseases. *Acinetobacter baumannii* can be seen as one of these resilient pathogens due to its increasing prevalence in hospitals, its resistance to treatment, and its association with elevated mortality rates. Despite its clinical significance, the scientific understanding of this pathogen in non-hospital settings remains limited. Knowledge of its virulence factors is also lacking. Therefore, in this review, we seek to shed light on the latest research regarding the ecological niches, microbiological traits, and antibiotic resistance profiles of *Acinetobacter baumannii*. Recent studies have revealed the presence of this bacterium in a growing range of environmental niches, including rivers, treatment plants, and soils. It has also been discovered in diverse food sources such as meat and vegetables, as well as in farm animals and household pets such as dogs and cats. This broader presence of *Acinetobacter baumannii*, i.e., outside of hospital environments, indicates a significant risk of environmental contamination. As a result, greater levels of awareness and new preventive measures should be promoted to address this potential threat to public health.

## 1. Introduction

Today, antibiotic resistance presents a formidable global health challenge, as evidenced by World Health Organization (WHO) records, which indicate that approximately 700,000 individuals annually succumb to infections related to antibiotic resistance. Moreover, projections suggest that, without significant intervention, this issue could plunge 24 million people into extreme poverty by 2030 and result in a staggering 10 million deaths annually by 2050 [1,2]. In recognition of the urgency of this situation, the WHO in 2017 issued a list of 56 bacteria for which novel therapeutic approaches are required due to their resistance to conventional antibiotics. These bacteria pose a grave threat to global health characterized by longer hospital stays, higher treatment expenses, and elevated mortality rates among hospitalized patients [3,4]. Among these pathogens, *Acinetobacter baumannii* stands out as an especially significant concern. Now classified as one of the most critical multidrug-resistant bacteria, *Acinetobacter baumannii* contributes to 7300 cases of infection and 500 deaths each year and ranks among the most serious public health concerns due to a lack of effective chemical therapies and diagnostic tools to mitigate its potential risks [5].

*Acinetobacter baumannii* presents a significant challenge in healthcare settings due to its remarkable ability to develop resistance to a broad spectrum of antibiotics. This resistance arises from a combination of intrinsic and acquired mechanisms, making the treatment of infections caused by this bacterium increasingly complex. It is known to cause severe and invasive nosocomial infections including ventilator-associated pneumonia, bloodstream infections, infections of the urinary tract, infections of the skin and soft tissues, and meningitis [6,7]. Numerous studies have identified hospital intensive care units, neonatal areas, and burns units [8,9] as the primary locations where critically ill patients acquire infections from this opportunistic pathogen. These environments create an optimal breeding ground for the rapid transmission of *Acinetobacter baumannii* among patient populations. In short, *Acinetobacter baumannii* is a potentially infectious emerging pathogen that requires targeted interventions and effective chemical therapies.

In the present study, we sought to analyze recent research on pathogens across different environmental niches and to highlight any significant findings. To this end, we curated and prioritized a selection of recently published papers that address the incidence and presence of *Acinetobacter baumannii* in out-of-hospital environments. By highlighting new niches that have facilitated its proliferation, we sought to shed light on the expanding scope of its environmental reservoirs.

## 2. Methodology

A search of the Scopus database was conducted for scientific articles published between 2017 and 2022, using a search equation with keywords such as “*Acinetobacter baumannii*”, “incidence”, and “antibiotic resistance”. The inclusion criteria covered published reports on the incidence or presence of the pathogen in out-of-hospital settings, as well as papers concerning the resistance profile and/or virulence genes of *Acinetobacter baumannii*. The exclusion criteria covered articles in languages other than Spanish or English, letters to editors, abstracts, clinical cases, commentaries, and memoirs. Three reviewers independently screened titles, abstracts, and keywords to manually select eligible articles. When papers lacked sufficient information in these three sections, but were still considered potentially suitable, the full text was consulted (Figure 1).

All papers were extracted and organized in a Microsoft Excel spreadsheet, with data for each document including the title, authors, year of publication, and abstract. This information was first used to facilitate a descriptive analysis. Subsequently, each reviewer ranked the articles in order of priority based on their stated goals. Finally, papers were collated and selected for a descriptive analysis. A priority-based system was employed for classification into four categories, as follows: very high priority (meeting all inclusion criteria and fully addressing the review’s objective); high priority (meeting inclusion criteria and containing relevant information, though not fully addressing the objective); medium priority (meeting inclusion criteria, but with insufficient information to fully address the objective); and low priority (meeting criteria, but containing irrelevant information). Discrepancies regarding inclusion were addressed through dialogue and resolved accordingly [10,11]. Papers classified as being of very high or high priority were used for the subsequent in-depth analysis.

## 3. Results and Discussion

From the Scopus database search, a total of 545 documents were initially retrieved. After applying the first filter, 87.15% (475 out of 545) of the documents met the inclusion criteria (Figure 1). Applying the “priority criteria” to the selected documents, it was determined that 47.36% (225 out of 475) of the documents fell into the low-priority category, while 28.0% (133 out of 475) were classified as being of medium priority. Only 24.63% (117 out of 475) of the documents were deemed to be of high or very high priority, consisting of 84 and 33 papers, respectively. Works categorized as being of low or medium priority (75.37%) were excluded due to their focus on issues not directly relevant to the study of the opportunist pathogen; such issues included mechanisms of pathogenicity, epidemiology, incidence in hospitals, and resistance profiles in clinical samples, among others. Finally, documents classified as being of very high or high priority (117) were included in the in-depth analysis and thus provided the information described below.

The Vos Viewer keyword analysis unveiled a notable network of interactions comprising 229 interconnected keywords, each appearing at least three times, linked by a total of 2359 interactions and a cumulative link strength of 68,690 (Figure 2). These keywords were classified into five distinct clusters, with the largest three clusters centering on terms such as “*Acinetobacter baumannii*”, “antimicrobial resistance”, and “antibiotic agent”, along with other pertinent terms related to the biochemistry and genetics of the bacteria. Additionally, there were peripheral nodes with limited interactions within the network, including keywords such as “water”, “environment”, “wastewater”, and “non-human”. This last observation suggests that such areas of study need for more attention in the literature.

### 3.1. Microbiological Characteristics and Virulence Factors of Acinetobacter baumannii

*Acinetobacter baumannii* is categorized as an emerging pathogen within the Moraxellaceae family and the *Acinetobacter genus*. It is characterized as a Gram-negative coccobacillus and it exhibits several distinctive microbiological features, including non-motility, positive catalase activity, negative oxidase activity, strict aerobic metabolism, an absence of indole production, non-fermentative behavior, and a lack of spore formation. When cultured at 37 °C, it forms mucoid, smooth, grayish-white colonies on solid media [12,13].

Despite its relatively recent recognition, *Acinetobacter baumannii* has swiftly garnered global attention due to its transition from a low-grade pathogen to a leading cause of mortality and morbidity in hospital-acquired infections, particularly pneumonia. Notably, it exhibits a broad resistance profile against various antimicrobial agents, and its repertoire of virulence factors enables the successful colonization of hosts. Moreover, its remarkable adaptability facilitates rapid mutation to suit its requirements within a given environment. Consequently, *Acinetobacter baumannii* poses a significant and escalating threat to public health on a global scale.

Table 1 lists the main virulence factors used by *Acinetobacter baumannii* to colonize a host, cause disease, evade the immune response to survive, or transmit to a new host; each virulence factor has a specific function that is used at different stages of its life cycle. Various genomic analyses, phenotypic analyses, and infection models have enabled researchers to determine that *Acinetobacter baumannii* contains 16 gene islands involved in virulence [14]; this means that the pathogen has a high number of known virulence factors and perhaps more that remain unknown at present.

Biofilm formation stands out as the most critical virulence factor expressed by *Acinetobacter baumannii*, as it facilitates host-cell colonization and promotes the evasion of antibiotic action, ultimately leading to multidrug resistance. This characteristic significantly heightens the likelihood of patient mortality, as effective chemical therapies to combat this nosocomial pathogen are lacking. The presence and persistence of biofilms in clinical environments provide favorable conditions for proximity to potential hosts, colonization, and infection. Moreover, biofilms confer resistance to disinfectants, increase survivability in hostile environments, and enable adherence to hospital surfaces, including medical devices, such as ventilators and stethoscopes, where the pathogen is commonly found [15,16].

**Table 1 microorganisms-12-00644-t001:** Main virulence factors of *Acinetobacter baumannii*.

Virulence Factor	Activity in Pathogenesis	Reference
Outer-membrane proteins: (OmpA, Omp 33–36, Omp 22)	Adherence and invasion, apoptosis induction, biofilm formation, persistence, and serum resistance.	[17,18]
Lipopolysaccharide (LPS)	Serum resistance, survival during tissue infection, evasion of immune system response.	[18,19]
Micronutrient Acquisition System (iron, manganese and zinc)	Competition and survival cause the death of host cells.	[20]
Capsular polysaccharide	Serum resistance, survival during tissue infection, evasion of antimicrobial action.	[18,19]
Phospholipases (PLC and PLD)	Serum resistance, cytolytic activity, invasion, in vivo survival.	[18]
Secretion system type II, V and VI	Adherence and colonization, in vivo survival, bacterial competition, biofilm formation.	[18,19,21]
Biofilm	Adherence, survival in environments, and resistance to antibiotics.	[18,21,22]
Quorum sensing	Biofilm formation.	[17]
Pili type IV	Motility, adherence, biofilm formation.	[13,17]
Outer-membrane vesicles	Release of virulence factors, horizontal transfer of antibiotic resistance genes.	[17,21]

### 3.2. Natural Reservoirs of Acinetobacter baumannii

The genus Acinetobacter encompasses several species characterized as free-living saprophytes; these are widely distributed in various environmental niches including water, soil, food, wastewater, animals, and humans [23]. Among these species, *Acinetobacter baumannii*, Acinetobacter haemolyticus, and Acinetobacter calcoaceticus are of clinical significance [24]. Initially identified as a species in 1986, *A. baumannii* has gained prominence in recent decades as a key causative agent of healthcare-associated infections (HAIs), particularly affecting patients who require prolonged hospital stays [17]. Consequently, research on *A. baumannii* has predominantly focused on hospital epidemiology, specifically in intensive care and burns units. However, there have been far fewer studies of its incidence in the environment and animal healthcare settings, hindering the identification of its out-of-hospital origins.

### 3.3. Hospital Habitats of Acinetobacter baumannii

The incidence and resistance of *Acinetobacter baumannii* have both been frequently documented in hospital settings. It has been isolated in intensive care units, and it has also been detected on reusable medical equipment such as ventilators, humidifiers, blood pressure monitoring devices, stethoscopes, laryngoscopes, curtains, mattresses, pillows, urinals, sinks, door handles, and even the gowns and gloves of hospital staff. It has also been detected in bronchial and oropharyngeal secretions, as well as in the digestive tract [24,25].

Its spread and prevalence within healthcare institutions are facilitated by its ability to survive in both dry and humid environments, being able to withstand such conditions for periods between 25 and 100 days. Additionally, the resistance of *Acinetobacter baumannii* to disinfectants and antibiotics, coupled with its biofilm-forming capability, enables it to colonize inert surfaces and medical devices. This feature is attributed to the presence of capsular polysaccharides, which allow the bacterium to thrive in nutrient-limiting conditions [26,27,28].

### 3.4. Out-of-Hospital Habitats for Acinetobacter baumannii

Reports of community-acquired *A. baumannii* infections in subtropical and tropical areas suggest that the pathogen may be acquired from sources outside of hospitals. In light of such findings, environmental isolates might be considered epidemiologically important in all countries [29]. However, the existence of out-of-hospital reservoirs of *A. baumannii* remains controversial. Today, there is still a lack of clear evidence regarding the natural habitat of the pathogen outside hospitals, its introduction to hospitals, and any subsequent return to the natural environment. Several studies have documented isolations of *A. baumannii* from various out-of-hospital habitats, including water bodies [30,31], soil [32,33], food [34,35], meat from food animals [36,37], vegetables [38,39], farm animals [40], domestic animals (cats [41,42], dogs [43,44], and horses [45]), wild animals [46,47], and even body lice [48].

It is noteworthy that the scientific literature indicates a nascent effort to elucidate the mysteries surrounding *A. baumannii* in out-of-hospital environments. While such investigations constituted only 24.63% of papers (117 out of 475) in the present review, compared with the 75.36% (358 out of 475) of studies that were conducted in hospitals, the publication trend suggests a rapid increase in information concerning the presence and behavior of *A. baumannii* outside of hospital settings. This growing interest reflects a recognition of the importance of understanding the broader ecological context and transmission dynamics of the pathogen. As research in this area expands, it is anticipated that our understanding of the epidemiology and risk factors associated with community-acquired *A. baumannii* infections will improve significantly. This trend underscores the importance of continued efforts to investigate and address the challenges posed by this resilient pathogen in both healthcare and community settings.

#### 3.4.1. Water Bodies

Water bodies may be used for recreational purposes or the irrigation of food crops for human consumption, amongst many other uses. However, if water bodies are not adequately treated to reduce contamination levels, they can pose serious threats to human and environmental health. Opportunistic biofilm-forming pathogens like *Acinetobacter* spp., carrying multiple antibiotic resistance genes, have been identified in wastewater and in natural aquatic environments [49,50].

For example, one study conducted at the Mthatha Dam in South Africa revealed the presence of different Acinetobacter species, with *A. baumannii* predominating, comprising 98 isolates, of which 53.1% (52 out of 98) exhibited resistance to six antibiotics [26]. Similarly, in Croatia, 10 strains of *A. baumannii* resistant to carbapenems and other antimicrobials were recovered from hospital wastewater. Such findings highlight the potential risk posed by untreated hospital wastewater as a focus for the proliferation of infections [51]. In another study, 26 *Acinetobacter* spp. strains were isolated from treated wastewater from a hospital in Mexico City. Five of these strains (19.2%) were resistant to amikacin, indicating their ability to withstand various wastewater treatment steps [30]. Considering that treated wastewater is now extensively used for crop irrigation, the further dissemination of the pathogen into new environmental niches such as soil and vegetation may be anticipated.

The presence of this opportunistic pathogen in aquatic environments also poses a risk to public health, particularly through horizontal gene transfer and potential interactions with other organisms consumed by humans, such as fish. By such means, aquatic environments may serve as a vehicle for spread into the community [26]. These findings underscore the importance of effective wastewater treatment measures and surveillance strategies to mitigate the transmission and spread of antibiotic-resistant pathogens in environmental and community settings.

Despite the evidence indicating the presence and persistence of *A. baumannii* in water bodies, it remains unclear whether water serves as a natural habitat or acts merely as a reservoir for this opportunistic pathogen. The question of whether its presence in water derives primarily from human contamination or animal sources has yet to be definitively answered.

#### 3.4.2. Soil, Vegetables, and Food

The prevalence of *A. baumannii* in soil has been reported in various regions, indicating its widespread distribution and persistence in environmental habitats. In the Eastern Cape province of South Africa, *A. baumannii* was detected in soil samples with a prevalence of up to 41% [52]. Similarly, Suresh et al. [33] identified *A. baumannii* strains in soil samples from Mangaluru, India, and found that they exhibit resistance to multiple drugs including fluoroquinolones, aminoglycosides, sulfonamides, tetracyclines, and carbapenems. *A. baumannii* has also been found in food items, highlighting potential routes of transmission to humans. Kanafani and Kanj [53] reported the presence of the pathogen in fruits, raw vegetables, milk, and dairy products. Additionally, on a farm in Japan, isolates of *A. baumannii* were identified in fresh vegetables [36]. Carvalheira et al. [27] detected eighteen species of Acinetobacter in lettuces and fruits (apples, pears, bananas, and strawberries); 11% of the strains belonged to *Acinetobacter baumannii*, 19.8% were classified as multidrug-resistant, and 4.4% were classified as being extensively drug-resistant. Another study evaluated the presence of *Acinetobacter* spp. resistant to β-lactams and cefotaxime in samples of commercially produced ready-to-eat salad; though *A. baumannii* was not identified in this work, the presence of other Acinetobacter species involved in human infections was elsewhere reported [54].

The study conducted by Bitar et al. [34] cast light on the potential role played by the international food trade in the dissemination of *Acinetobacter baumannii*-carrying resistance genes; these include colistin resistance genes (*mcr-4.3*), which are typically associated with hospital isolates. Such findings underscore the importance of considering the international food trade as a possible route for the spread of antibiotic-resistant pathogens within the human community.

Indeed, numerous investigations have highlighted the presence of various resistant pathogens in soil, plants, and food, with the implication that international trade is responsible for their transmission. The detection of *A. baumannii* in raw food imported into the Czech Republic for commercial use exemplifies this phenomenon. However, there is still much more that needs to be understood about the environmental survival of these pathogens, their competitiveness against other environmental strains, the mechanisms driving resistance selection, and the alternative pathways through which they might reach human populations [55].

#### 3.4.3. Products of Animal Origin for Human Consumption

The meat of animal origin, being rich in protein, provides an ideal environment for the growth of microorganisms during decomposition, including various species of Acinetobacter [46]. These bacteria are saprophytic and exhibit a high capacity to survive in diverse environments, including raw meat. In a study conducted in Buraydah City, Saudi Arabia, Elbehiry et al. [36] identified 55 strains of *A. baumannii* in 220 samples of raw meat sourced from camels, cows, sheep, and chickens. The incidence was notably higher in sheep (46.5%) and chickens (32.5%).

Similarly, in China, Tavakol et al. [37] reported the isolation of *A. baumannii* from meat intended for human consumption, with 22 strains detected in 126 samples. The incidence rates were 45.45% for chicken and 18.18% for cattle. Furthermore, Ghaffoori et al. [40] found that *A. baumannii* isolates resistant to tetracycline and cefoxitin were present in samples of raw turkey and chicken meat. These resistant strains were present in 20% of the total samples analyzed (200 out of 1000).

Table 2 lists recent studies in which meat for human consumption obtained from various animals was reported to be a probable means of transmission of *A. baumannii*. Several carbapenem-resistant isolates from skin and feces samples of pigs and cattle slaughtered for human consumption have recently been reported [44,46]. In addition, in isolates from raw sheep meat, most strains showed resistance to amoxicillin/clavulanic acid, gentamicin, tetracycline, ampicillin, and tobramycin [36].

The use of chemicals, including growth promoters and antibiotics, in animal husbandry may contribute to the emergence and spread of antibiotic-resistant bacteria. These chemicals are commonly used to enhance growth rates in livestock and to control infectious diseases within animal populations [55].

#### 3.4.4. Domestic Animals

Since 2011, *A. baumannii* has emerged as a significant pathogen in veterinary clinics, with isolations being reported from various sites; these include urinary tract infections, wound infections, abscesses, skin lesions, pus, manure, and catheters used in sick animals. Infections associated with *A. baumannii* in animals include otitis, canine pyoderma, pneumonia, mastitis, bronchopneumonia, feline necrotizing fasciitis, and sepsis, indicating a broad range of clinical manifestations [37,58]. Domestic animals and pets have been identified as potential reservoirs of the pathogen, contributing to its persistence and transmission within veterinary settings. (Table 2). Dogs, cats, and horses are among the animals most frequently hospitalized due to diseases associated with *A. baumannii* [59].

There have also been reports of *A. baumannii* isolates from healthy canine skin [60]. The authors of [61] identified this pathogen, with different resistance patterns, in 23 of 80 surface isolates from the skin, feces, urine, and ecdysis remains of domestic reptiles imported from different countries to Frankfurt airport, Germany. Although the bacterial genus *Acinetobacter* spp. is non-pathogenic in reptiles, the authors suggested that a deeper understanding of the molecular epidemiology of the pathogen is needed [61].

Also in Germany, Wareth [45] isolated an *A. baumannii* strain from a horse with conjunctivitis and determined the sequence of its genome and its gene distribution, including resistance genes. They suggested that further studies be conducted to assess the possible zoonotic risk of this agent for human and animal health.

#### 3.4.5. Other Sites

The epidemiology of healthcare-associated infections and the surveillance of causative pathogens such as *Acinetobacter baumannii* both emphasize the importance of investigating the potential airborne spread of the pathogen within hospital environments. Isolates of *A. baumannii* have been reported from various hospital surfaces, including pillows and sheets, indicating the potential for environmental contamination [44]. In a Danish study, *A. calcoaceticus* and *A. baumannii* were recovered from an intensive care unit using an environmental sampler with sedimentation plates, highlighting the presence of these bacteria in the hospital air [62]. The airborne transmission of *A. baumannii* within hospital settings remains an area requiring further investigation; however, it is plausible that air in both closed and open spaces could serve as a vehicle for the dissemination of the pathogen. In Jordan, Ababneh [63] isolated 63 strains of *A. baumannii* from floors and high-contact surfaces in various urban settings, including university campuses, shopping malls, parks, playgrounds, markets, and ATMs. Sequencing studies revealed that 39.6% of the isolated bacteria were new strains, most of which could form biofilms. Furthermore, four strains were identified as being multidrug-resistant [63].

As previously mentioned, pathogenic microorganisms from infected patients can spread by various means inside hospitals (Figure 3). Coughing, sneezing, and even talking can propel infectious particles into the air, leading to airborne transmission. Improper disposal of infectious materials can contaminate surfaces like doorknobs, countertops, medical equipment, and patient bedding, creating risks for both healthcare workers and patients. Handwashing, showering, and flushing of toilets can all introduce pathogens into wastewater systems, potentially contributing to waterborne transmission. Additionally, insects or rodents can act as vectors, spreading pathogens to different areas within hospitals. Finally, human-to-human transmission can occur unknowingly through contact between healthcare workers, patients, and visitors.

These pathogens can also escape the hospital environment and disperse further. Hospital visitors and workers can carry pathogens to their homes, workplaces, or community settings, extending the spread. Contaminated items like laundry, equipment for repair, or waste disposed of in landfills can introduce pathogens to new environments. Untreated wastewater or sewage can contaminate water bodies or soil, creating a significant environmental risk. Inappropriately treated wastewater used for irrigation and un-inactivated sludge applied as a soil amendment can both lead to crop contamination. Ingesting microbially contaminated crops can pose a risk of infection for animals and people. Hospital air currents can carry pathogens outdoors, potentially affecting nearby areas through airborne transmission. In some cases, insects infected within the hospital can act as vectors, spreading pathogens to nearby food sources, animals, or surfaces. Pathogenic bacteria may also arise from diverse sources outside hospitals, and all such bacteria represent distinct risks to public health. A primary reservoir lies within the food chain. Human and animal excrement, water and soil polluted due to agricultural runoff, and animals both domestic and wild all may serve as vectors for pathogenic bacteria. In addition to environmental issues, social determinants such as population density, socioeconomic status, and access to healthcare all intersect to shape the prevalence and transmission dynamics of bacterial pathogens.

In the case of *A. baumannii,* its wide distribution in various environmental niches, including water bodies, soil, and food, suggests potential natural reservoirs beyond human activities. The environmental distribution of *A. baumannii* can be explained by its adaptability and ability to survive in diverse ecological niches. However, this also complicates the determination of its origin. Although it does appear that hospital environments represent the main source, the genetic diversity of *A. baumannii* strains may reflect multiple routes of transmission and sources of contamination. While human activities such as wastewater discharge and agricultural runoff likely contribute to the presence of *A. baumannii* in water bodies, the potential role of animal reservoirs cannot be overlooked. To elucidate the natural habitat or reservoirs of *A. baumannii* in water bodies, comprehensive studies integrating molecular epidemiology, ecological surveys, and comparative genomics are warranted. In addition, the presence of this opportunistic pathogen in agricultural settings and food products raises concerns about potential transmission to humans through consumption. Although the precise routes of dissemination of *A. baumannii* from animals or vegetables to humans are not yet fully understood, several pathways have been proposed. These include direct contact with contaminated animals/vegetables or their environments, the consumption of contaminated food products, and exposure to contaminated water sources. These proposed transmission routes all emphasize the need for comprehensive surveillance, risk assessment, and control strategies along the food supply chain to mitigate the spread of opportunistic pathogens through international trade.

More investigations are needed to unravel the complex dynamics of *A. baumannii* transmission in environmental settings and to inform targeted interventions to mitigate its spread and impact on public health. Efforts to implement stringent hygiene practices throughout the whole process of meat production and supply are essential to mitigate this risk and safeguard public health.

During the preparation of this work, a preprint was identified with interesting and notable results. The extensive study examined *A. baumannii* in over 1300 white stork nestlings across Poland and Germany, revealing its presence in soil, plant roots, and, potentially, earthworms. By identifying soil and compost as hotspots, researchers tracked A. baumannii’s year-round population dynamics. The bacteria’s rapid colonization of sterilized plant material suggests airborne dispersal and a “patrolling” behavior for new habitats. Linking this to fungal interactions and high mortality in stork nestlings, the study proposes a long coevolution between *A. baumannii* and fungi. This coevolution might explain *A. baumannii’s* intrinsic antibiotic resistance and its ability to adhere to and suppress fungal spores. Furthermore, the bacterium’s resistance to stress might be an adaptation for intercontinental travel, hitching rides on fungal spores [64]

### 3.5. Antimicrobial Resistance of A. baumannii from Out-of-Hospital Isolates

Chromosomal beta-lactamase genes are the primary example of natural resistance genes in *A. baumannii* because they offer some baseline defense against β-lactam antibiotics. AmpC β-lactamase and OXA-51-type genes are important factors in the natural resistance to lactamases in *A. baumannii*. Both offer limited intrinsic resistance, primarily affecting penicillin, but their activity against other β-lactam classes is usually weak. The true concern lies in their potential to contribute to a broader resistance profile when combined with other mechanisms.

While chromosomal β-lactamases provide a baseline defense for *A. baumannii*, the true danger of *A. baumannii*’s resistance profile stems from its ability to acquire genes encoding potent β-lactamases on plasmids. These mobile genetic elements facilitate the rapid spread of resistance. Two prominent examples highlight this threat: Extended-Spectrum Beta-Lactamase (ESBL) genes encode enzymes that can particularly hydrolyze cephalosporins. OXA-23 and OXA-48 genes encode carbapenemase enzymes, conferring resistance to even the most powerful carbapenems, a last-resort class of antibiotics for multidrug-resistant infections. Notably, OXA-23 can be chromosomally or plasmid-mediated, while OXA-48 is typically found on plasmids.

Table 3 exemplifies the spectrum of resistance genes harbored by *A. baumannii* strains isolated from various hospital and non-hospital settings. In hospital environments, class A and class B β-lactamase genes (including IMP, VIM, SIM, AmpC, SPM, and NDM-1) are frequently observed [65]. Additionally, these strains often exhibit resistance to aminoglycosides via strA and srtB genes.

In contrast, the prevalence of resistance genes appears lower in *A. baumannii* isolates from non-hospital environments. However, a study in Germany reported a concerning trend: isolates from powdered milk displayed universally high resistance to chloramphenicol and oxacillin [35]. Resistance to other antibiotics like cefotaxime and cefepime has also been documented, albeit less frequently. Notably, these non-hospital isolates often retain susceptibility to tetracycline, tobramycin, erythromycin, and ciprofloxacin.

Beyond the hospital, *A. baumannii* exhibits remarkable ecological versatility, colonizing diverse niches that include soil, water, animals, and even humans. These bacteria also demonstrate exceptional resilience in harsh environments. This adaptability, coupled with the ease of horizontal gene transfer through plasmids, allows *A. baumannii* to become resistant to a vast array of antibiotics. The combined presence of natural and acquired resistance mechanisms elevates *A. baumannii* to a serious public health threat.

These findings suggest variations in the resistance profiles of *A. baumannii* isolates between hospital and out-of-hospital environments, and thus highlight the importance of understanding and monitoring antimicrobial resistance in diverse ecological niches to inform effective control measures and antimicrobial stewardship practices.

Acinetobacter species within the *A. calcoaceticus*–*A. baumannii* (Acb) complex are clinically important but difficult to differentiate due to their high genotypic diversity and similar phenotypic characteristics. Multilocus Sequence Typing (MLST), following either the Oxford or Pasteur scheme (or sometimes both), is a common method for assigning sequence types (STs) to these species [59]. This study analyzed various methodologies used in the literature for *A. baumannii* detection, ranging from basic microbiology tests and CHROMagar Acinetobacter to molecular typing methods like MLST. Notably, MLST has been applied to identify *A. baumannii* in both hospital and out-of-hospital settings.

Molecular epidemiology has revealed ST1, ST2, and ST3 as the major clonal lineages within *A. baumannii* populations. Subsequent research has identified additional epidemic lineages, providing valuable insights into the distribution, spread, and biological characteristics of these clonal groups.

Table 4 summarizes the STs reported in the analyzed documents for *A. baumannii* isolated from both hospital and out-of-hospital environments. Notably, the table highlights the presence of STs beyond the major clonal lineages, including ST10, ST12, ST20, ST25, ST46, ST49, ST162, and others. This diversity suggests a broader genotypic and phenotypic range within *A. baumannii*, potentially contributing to the global spread of successful lineages.

## 4. Conclusions

The presence of numerous virulence factors expressed by *A. baumannii* underscores its emergence as a pathogen that is potentially dangerous to public health. Its capacity to acquire new virulence factors signifies an evolutionary advance in its pathogenicity, which enables it to develop novel strategies for the adhesion, colonization, and invasion of host cells. Biofilm formation further enhances its adaptability by facilitating adherence to various surfaces, protecting against adverse conditions, and promoting its survival in diverse habitats.

The acquisition of new resistance genes further exacerbates the threat posed by *A. baumannii*, rendering it a latent risk to human health. Its ability to evade the action of multiple antibiotics is particularly concerning given the lack of efficient therapeutic options to combat infectious outbreaks caused by this pathogen.

In this paper, we summarized various recent studies that shed light on the success of *A. baumannii* in adapting to different habitats and potentially serving as important out-of-hospital reservoirs. These findings underscore the real risk posed by antibiotic-resistant pathogens such as *A. baumannii* in soil, plants, and food, which often go unnoticed in these environmental niches.

In conclusion, understanding the ecological dynamics and transmission pathways of *A. baumannii* in diverse habitats is crucial for devising effective strategies to mitigate its spread and minimize its impact on public health. Continued research and surveillance efforts are essential to monitor and address the evolving threat posed by this opportunistic pathogen.

## Figures and Tables

**Figure 1 microorganisms-12-00644-f001:**
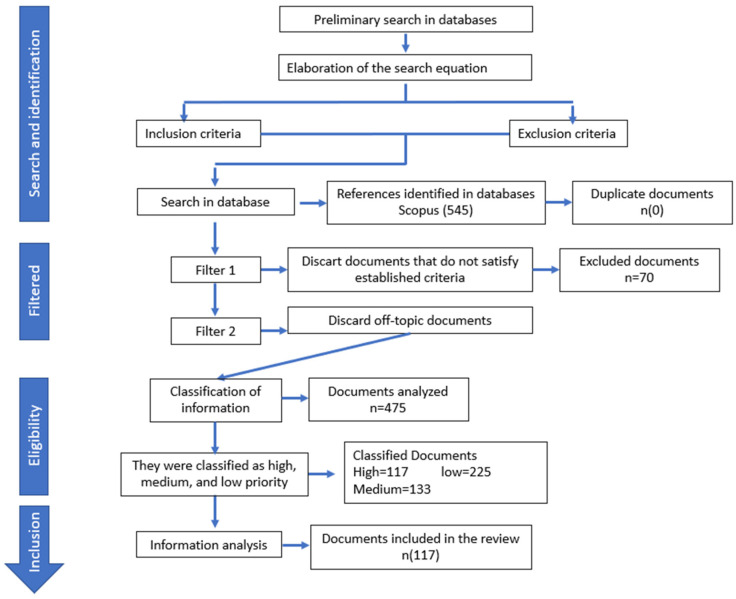
Document collection and analysis process.

**Figure 2 microorganisms-12-00644-f002:**
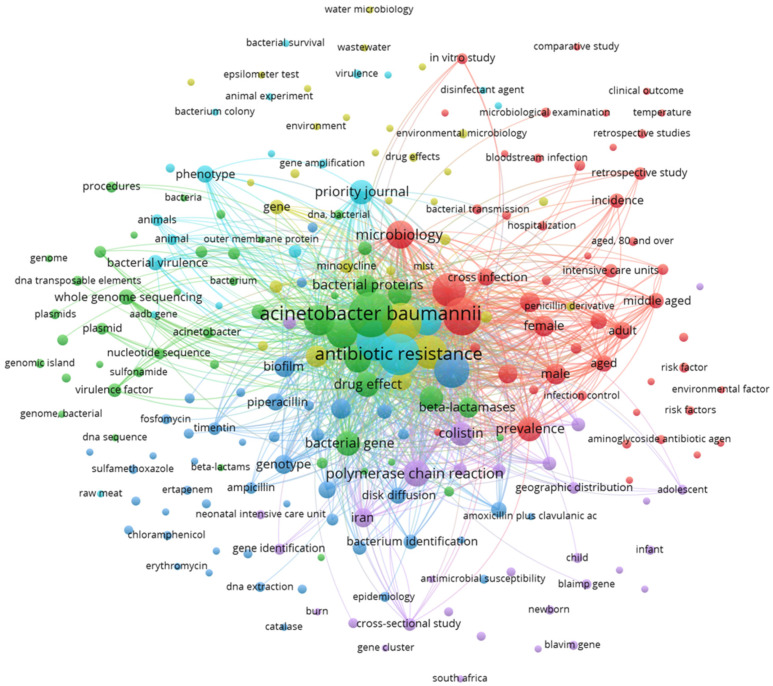
Network analysis of keywords with at least three occurrences. VOSviewer version 1.6.20. Copyright 2009–2023 Nees and Jan van Eck and Ludo Waltman.

**Figure 3 microorganisms-12-00644-f003:**
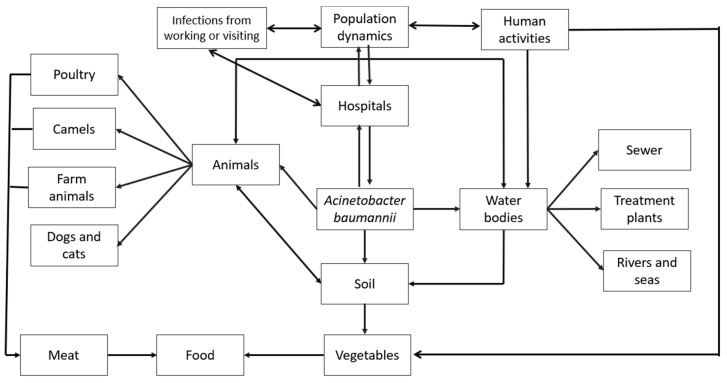
Environmental distribution of *A. baumannii* in out-of-hospital habitats.

**Table 2 microorganisms-12-00644-t002:** Isolation of *Acinetobacter baumannii* in animals.

Animals for Consumption	Isolations	Reference
Sheep	Meat	[36,44,46]
Goat	Meat	[44,46]
Cow	Meat	[44,46,56]
Camels	Meat	[36,37,44,46]
Pork	Meat, manure	[29,44,57]
Chicken/turkey	Meat	[36,37,40]
Cats	Wounds, pus, urine, skin	[44,58]
Dog	Wounds, pus, urine, skin	[44,58]
Turkey	Meat	[37,44]
Horse	Wounds, pus, catheter	[37,44]

**Table 3 microorganisms-12-00644-t003:** Antibiotic resistance genes reported for *A. baumannii*.

Gene Group	Gene	Isolates	Reference
β-lactamases (class A)	Penicillins: *Tem-1*, *Tem-2* Carbenicillinase: *CARB-5* *BLEE: VEB-1*, *PER-1*, *SHV*, *TEM-92* and *CTX-M-2* Carbapenems: *oxa23*, *oxa24*, *oxa 40*, *oxa 48*, *oxa 51*, blaKPC	Environmental and hospital, COVID patient samples, lettuce, slaughterhouses, water, air, water, animal fecal matter, dog skin, dog and cat urine, turkey and chicken meat	[26,40,43,50,52,54,59,60,65,66,67,68,69,70,71,72]
β-lactamases (class B)	Metallo-β-lactamase (MBL): *IMP*, *VIM*, *SIM*, *SPM* and *NDM-1* Streptomycin *strA* and *srtB*	Environmental and hospital samples, slaughterhouses, and water	[26,35,50,59,65,69,73]
β-lactamases (class C)	Bla *AmpC*, *blaACC*, *blaDHA*	Clinical samples, lettuce	[54,59]
β-lactamases (class D)	Oxacillinases: *oxa 58*, *oxa 143*	Dog and cat samples, clinical samples	[26,40,59]
Quinolone	*abaQ*	Clinical samples	[74]
Sulfonamide	*sul1* *sul2*	Clinical samples	[75]
Tetracyclines	*Tet (A)* *Tet (B)* *Tet39*	Clinical samples, water and soil, turkey and chicken meat	[40,52,75,76]

**Table 4 microorganisms-12-00644-t004:** Common reported Multilocus Sequence Typing (MLST).

Multilocus Sequence Typing	Nomenclature	Source	References
ST-455, ST1293, ST1296, ST11ST14, ST368, ST1298, ST195, ST1295	Oxford scheme	Hospital	[6]
ST1, ST2, ST25, ST85, and ST215	Pasteur scheme	Clinic ST1 and ST2 out of the hospital	[66]
ST2, ST16, ST23, ST1406, and ST1407	Pasteur scheme	Hospital	[73]
ST2, ST281, ST25, and ST406	Pasteur scheme	Hospital	[70]
ST1584, ST1607, ST1608, ST1609, ST1610, ST1611, and ST1612	Oxford scheme	Soil sample	[32]
ST2 (Pasteur’s scheme) ST451 (Oxford’s scheme)	Pasteur and Oxford scheme	Dog and cat feces	[59]
ST-195	Oxford scheme	Pig manure from a farm	[29]
ST149, ST164, ST25 (human), ST203, and ST1198	Pasteur scheme	Hospitalized dogs and cats	[42]
ST1027 ST155, ST80, ST504, ST690, ST402, ST1, ST2, and ST25	Pasteur scheme	Farm bovines	[56]
ST 162, ST 1014, and ST492	Pasteur scheme	Cow and pig	[57]
ST25 (human) ST195 (mature)		It mentions that the same ST can be found in different species such as humans, pigs, cows, etc.	[44]
ST25, ST46, ST49, ST220, and ST249	Pasteur scheme	Reptiles	[61]
ST1 (cat) ST2 (dogs, cattle) ST10 (dog) ST12 (cat) ST25 (cat) ST491 (pig) ST20, ST492, and ST493 (poultry)	Pasteur scheme	Different animals	[58]
ST46	Pasteur scheme	Horse	[45]

## Data Availability

No new data were created or analyzed in this study. Data sharing is not applicable to this article.

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
