# Peer review of "Acinetobacter baumannii, a Multidrug-Resistant Opportunistic Pathogen in New Habitats: A Systematic Review"

_microorganisms, 2024, doi:10.3390/microorganisms12040644_

Round 1
Reviewer 1 Report
Comments and Suggestions for Authors
1、It is suggested that the number of keyword occurrences can be increased appropriately when plotting the interaction network, thus increasing significance.
2、It is recommended that the discussion of the transmission of Acinetobacter baumannii from the nosocomial to the extranosocial environment be appropriately added.
3、Some references have too many authors, preferably no more than three.
4、In line 284, "mcr-4.3" should be italicized.
5、Data on pathogens of fruit origin should be collected.
6、In table 2 & 4, A lot of formatting needs to be changed.
Comments on the Quality of English LanguageA lot of language modification is required.
Author Response
We are grateful for all the reviewers' comments; we have considered all of them and made the corresponding corrections to improve the quality and readability of the manuscript. In the following, we respond point by point to the suggestions, observations, and corrections made.
Reviewer 1
R1. It is suggested that the number of keyword occurrences can be increased appropriately when plotting the interaction network, thus increasing significance.
AU: Thank you for your valuable suggestion. Reducing the frequency of keywords can indeed enhance the overall significance. However, in this work, we aimed to emphasize the prevalence of keywords associated with out-of-hospital habitats. While we only saw a few interactions, this field is clearly gaining interest. More research is needed though, to fully understand the complexities and improve our grasp of the issue. Therefore, we keep the same keyword occurrences to emphasize the message. Hope the reviewer agrees with this last statement.
R1. It is recommended that the discussion of the transmission of Acinetobacter baumannii from the nosocomial to the extra social environment be appropriately added.
AU: Thanks for your suggestion. To address it, we've added a new section that explains the general environmental distribution of pathogenic microorganisms. We then specifically focus on A. baumannii, highlighting its presence in various environmental settings. However, there's still a lot to learn about how exactly these bacteria spread in the environment:
As previously mentioned, pathogenic microorganisms from infected patients can spread by various means inside hospitals (Figure 3). Coughing, sneezing, and even talking can propel infectious particles into the air, leading to airborne transmission. Improper disposal of infectious materials can contaminate surfaces like doorknobs, countertops, medical equipment, and patient bedding, creating risks for both healthcare workers and patients. Handwashing, showering, and flushing of toilets can all introduce pathogens into wastewater systems, potentially contributing to water-borne transmission. Additionally, insects or rodents can act as vectors, spreading pathogens to different areas within hospitals. Finally, human-to-human transmission can occur unknowingly through contact between healthcare workers, patients, and visitors.
These pathogens can also escape the hospital environment and disperse further. Hospital visitors and workers can carry pathogens to their homes, workplaces, or community settings, extending the spread. Contaminated items like laundry, equipment for repair, or waste disposed of in landfills can introduce pathogens to new environments. Untreated wastewater or sewage can contaminate water bodies or soil, creating a significant environmental risk. Inappropriately treated wastewater used for irrigation and un-inactivated sludge applied as a soil amendment can both lead to crop contamination. Ingesting microbially contaminated crops can pose a risk of infection for animals and people. Hospital air currents can carry pathogens outdoors, potentially affecting nearby areas through airborne transmission. In some cases, insects infected within the hospital can act as vectors, spreading pathogens to nearby food sources, animals, or surfaces. Pathogenic bacteria may also arise from diverse sources outside hospitals, and all such bacteria represent distinct risks to public health. A primary reservoir lies within the food chain. Human and animal excrement, water and soil polluted due to agricultural runoff, and animals both domestic and wild all may serve as vectors for pathogenic bacteria. In addition to environmental issues, social determinants such as population density, socioeconomic status, and access to healthcare all intersect to shape the prevalence and transmission dynamics of bacterial pathogens.
In the case of A. baumannii, its wide distribution in various environmental niches, including water bodies, soil, and food, suggests potential natural reservoirs beyond human activities. The environmental distribution of A. baumannii can be explained by its adaptability and ability to survive in diverse ecological niches. However, this also complicates the determination of its origin. Although it does appear that hospital environments represent the main source, the genetic diversity of A. baumannii strains may reflect multiple routes of transmission and sources of contamination. While human activities such as wastewater discharge and agricultural runoff likely contribute to the presence of A. baumannii in water bodies, the potential role of animal reservoirs cannot be overlooked. To elucidate the natural habitat or reservoirs of A. baumannii in water bodies, comprehensive studies integrating molecular epidemiology, ecological surveys, and comparative genomics are warranted. In addition, the presence of this opportunistic pathogen in agricultural settings and food products raises concerns about potential transmission to humans through consumption. Although the precise routes of dissemination of A. baumannii from animals or vegetables to humans are not yet fully understood, several pathways have been proposed. These include direct contact with contaminated animals/vegetables or their environments, consumption of contaminated food products, and exposure to contaminated water sources. These proposed trans-mission routes all emphasize the need for comprehensive surveillance, risk assessment, and control strategies along the food supply chain to mitigate the spread of opportunistic pathogens through international trade.
More investigations are needed to unravel the complex dynamics of A. baumannii transmission in environmental settings and to inform targeted interventions to mitigate its spread and impact on public health. Efforts to implement stringent hygiene practices throughout the whole process of meat production and supply are essential to mitigate this risk and safeguard public health.
R1. Some references have too many authors, preferably no more than three.
AU: Thanks for pointing out that. We have checked and corrected the style of references according to the Editorial guidelines.
R1. In line 284, "mcr-4.3" should be italicized.
AU: Corrected according to your observation.
R1. Data on pathogens of fruit origin should be collected.
AU: Thanks for your suggestions. We have added this information as follows:
Carvalheira et al. (27) detected eighteen species of Acinetobacter in lettuces and fruits (apples, pears, bananas, and strawberries); 11% of the strains belonged to Acinetobacter baumannii, 19.8% were classified as multidrug, and 4.4 % were classified as being extensively drug-resistant. Another study evaluated the presence of Acinetobacter spp. resistant to b-lactams and cefotaxime in samples of commercially produced ready-to-eat salad; though A. baumannii was not identified in this work, the presence of other Acinetobacter species involved in human infections was elsewhere reported (54).
R1. In Tables 2 & 4, A lot of formatting needs to be changed.
AU: Thanks for your observation, we apologize for that, Tables were all checked and changed according to Editorial guidelines.
Reviewer 2 Report
Comments and Suggestions for Authors
The relevance of this review dedicated to ecology of Acinetobacter baumannii is beyond doubt. The authors conducted an in-depth analysis of modern literature. The data presented will contribute to a deeper understanding of the problem of distribution of Acinetobacter baumannii in the environment and stimulate new research in this direction. However, certain points require clarification and correction.
· Although the problem of antimicrobial resistance is not the main topic of the review, the authors need to approach the issue more carefully.
o It is necessary to distinguish between acquired and natural resistance (chromosomal beta-lactamases of the OXA-51 type).
o Lines 399 – 401. It is necessary to indicate more clearly which genes are responsible for resistance to carbapenems and aminoglycosides. Class C beta-lactamases (AmpC) do not usually cause resistance to carbapenems.
o Table 4 contains inaccuracies and errors:
§ class B beta-lactamase line includes aminoglycosides;
§ use "end" instead of "y";
§ decipher the abbreviation BLEE;
§ adequately place references to individual publications; it is not necessary to indicate the names of the authors.
Minor remarks.
· Check table numbers, number 1 is used twice.
· Line 215: “community-acquired” duplicated.
Suggestion for addition: a preprint was recently published: On the ecology of Acinetobacter baumannii - jet stream rider and opportunist by nature (DOI: 10.1101/2024.01.15.572815). Even though the work has not been peer-reviewed, given the importance of the results obtained, it seems to me appropriate to include it in the review.
Author Response
We are grateful for all the reviewers' comments; we have considered all of them and made the corresponding corrections to improve the quality and readability of the manuscript. In the following, we respond point by point to the suggestions, observations, and corrections made.
The relevance of this review dedicated to ecology of Acinetobacter baumannii is beyond doubt. The authors conducted an in-depth analysis of modern literature. The data presented will contribute to a deeper understanding of the problem of distribution of Acinetobacter baumannii in the environment and stimulate new research in this direction. However, certain points require clarification and correction.
R2. Although the problem of antimicrobial resistance is not the main topic of the review, the authors need to approach the issue more carefully.
AU: Thanks for your kind observation. You're right, the manuscript doesn't directly address A. baumannii's antimicrobial resistance, although it's certainly a relevant concern within the broader issue of this organism. We've reviewed the entire manuscript and made adjustments to ensure a clearer focus on the main topic.
R2. It is necessary to distinguish between acquired and natural resistance (chromosomal beta-lactamases of the OXA-51 type).
AU: Thanks for your observations. We have adjusted some parts of the text accordingly. For example, in the introduction section, we have added the next sentences:
Acinetobacter baumannii presents a significant challenge in healthcare settings due to its remarkable ability to develop resistance to a broad spectrum of antibiotics. This resistance arises from a combination of intrinsic and acquired mechanisms, making treatment of infections caused by this bacterium increasingly complex.
Also, in section 3.5 Antimicrobial resistance of A. baumannii from out-of-hospital isolates, the next paragraph was added:
Chromosomal beta-lactamase genes are the primary example of natural resistance genes in A. baumannii because they offer some baseline defense against -lactam anti-biotics. AmpC b-lactamase and OXA-51-type genes are important factors in the natural resistance to lactamases in A. baumannii. Both offer limited intrinsic resistance, primarily affecting penicillin, but their activity against other b-lactam classes is usually weak. The true concern lies in their potential to contribute to a broader resistance profile when combined with other mechanisms.
While chromosomal -lactamases provide a baseline defense for A. baumannii, the true danger of A. baumannii's resistance profile stems from its ability to acquire genes encoding potent b-lactamases on plasmids. These mobile genetic elements facilitate the rapid spread of resistance. Two prominent examples highlight this threat: Extend-ed-Spectrum Beta-Lactamase (ESBL) genes encode enzymes that can hydrolyze particularly cephalosporins. OXA-23 and OXA-48 genes encode carbapenemase enzymes, conferring resistance to even the most powerful carbapenems, a last-resort class of antibiotics for multidrug-resistant infections. Notably, OXA-23 can be chromosomally or plasmid-mediated, while OXA-48 is typically found on plasmids.
Table 3 exemplifies the spectrum of resistance genes harbored by A. baumannii strains isolated from various hospital and non-hospital settings. In hospital environments, Class A and Class B b-lactamases genes (including IMP, VIM, SIM, AmpC, SPM, and NDM-1) are frequently observed (64). Additionally, these strains often exhibit resistance to aminoglycosides via strA and srtB genes.
This adaptability, coupled with the ease of horizontal gene transfer through plasmids, allows A. baumannii to become resistant to a vast array of antibiotics. The combined presence of natural and acquired resistance mechanisms elevates A. baumannii to a serious public health threat.
R2. Lines 399 – 401. It is necessary to indicate more clearly which genes are responsible for resistance to carbapenems and aminoglycosides. Class C beta-lactamases (AmpC) do not usually cause resistance to carbapenems.
AU: thanks for your kind observation. We have re-writing the mentioned section as described in the previous replay.
R2. Table 4 contains inaccuracies and errors:
AU: We apologize for the mistakes. All tables were double-checked to avoid more mistakes.
R2. class B beta-lactamase line includes aminoglycosides;
AU: You are right. The table was corrected as well as the related sentences.
R2. use "end" instead of "y";
AU: Sorry for the mistakes. The entire manuscript was checked by experts from language editing services.
R2. decipher the abbreviation BLEE;
AU: Yes, sorry for that mistake. We corrected using the right abbreviation of Extended-Spectrum Beta-Lactamase (ESBL)
R2. adequately place references to individual publications; it is not necessary to indicate the names of the authors.
AU: Thanks for your observation. The entire manuscript was revised and corrected according to editorial guidelines.
Minor remarks.
R2. Check table numbers, number 1 is used twice.
AU: Again, sorry for the mistakes; we have corrected the numbers of tables and figures.
R2. Line 215: “community-acquired” duplicated.
AU: Tanks for the correction. Done
Suggestion for addition: a preprint was recently published: On the ecology of Acinetobacter baumannii - jet stream rider and opportunist by nature (DOI: 10.1101/2024.01.15.572815). Even though the work has not been peer-reviewed, given the importance of the results obtained, it seems to me appropriate to include it in the review.
AU: thanks for the recommendation. The information for the preprint was included as follows:
During the preparation of this work, a preprint was identified with interesting and notable results. The extensive study examined A. baumannii in over 1,300 white stork nestlings across Poland and Germany, revealing its presence in soil, plant roots, and potentially earthworms. By identifying soil and compost as hotspots, researchers tracked A. baumannii's year-round population dynamics. The bacteria's rapid colonization of sterilized plant material suggests airborne dispersal and a "patrolling" behavior for new habitats. Linking this to fungal interactions and high mortality in stork nestlings, the study proposes a long coevolution between A. baumannii and fungi. This coevolution might explain A. baumannii's intrinsic antibiotic resistance and its ability to adhere to and suppress fungal spores. Furthermore, the bacteria's resistance to stress might be an adaptation for intercontinental travel hitching rides on fungal spores (77).
Reviewer 3 Report
Comments and Suggestions for Authors
This manuscript summarizes the results of literature review about this important but less studied pathogen, Acinetobacter. This is timely and necessary review for the field but to deliver more useful information, the authors need to provide more comprehensive tables of summarized data )if necessary as supplemental file). The table should include prevalence of each AMR genes, virulence factors, biofilem formation, ST types of each habitat (for clinical isolates type of infection). For example, AMR genes including blaOXA and Sequence type ST2 are the some of the most prevalent AMR genes and St types among clinical isolates. This table will provide the status of this pathogens and the clues for their potential routes of transmission and contamination.
Minor points.
Figure 1; Discart > Discard
Line 152; host cell colonization?
Line 193-195; confusing sentence.
Line 215; delete one of the ‘community-acquired’
Line 285-287; confusing sentence.
Comments on the Quality of English Language
Minor editing of English language required. They are indicated in the minor points.
Author Response
We are grateful for all the reviewers' comments; we have considered all of them and made the corresponding corrections to improve the quality and readability of the manuscript. In the following, we respond point by point to the suggestions, observations, and corrections made.
This manuscript summarizes the results of literature review about this important but less studied pathogen, Acinetobacter. This is timely and necessary review for the field but to deliver more useful information, the authors need to provide more comprehensive tables of summarized data )if necessary as supplemental file). The table should include prevalence of each AMR genes, virulence factors, biofilem formation, ST types of each habitat (for clinical isolates type of infection). For example, AMR genes including blaOXA and Sequence type ST2 are the some of the most prevalent AMR genes and St types among clinical isolates. This table will provide the status of this pathogens and the clues for their potential routes of transmission and contamination.
AU: Thanks for your suggestions, very interesting and pertinent. Now, we have prepared four tables trying to provide sufficient information to know the environmental distribution of Acinetobacter baumannii; Analyzing these tables, alongside recent scientific literature, allows for the identification of potential environmental reservoirs for the bacteria. Focus on the following key points from the tables: the information includes the description of factors of virulence, the presence in environmental niches and biological samples; and the genes of resistance known. Despite the valuable data provided, significant knowledge gaps remain regarding A. baumannii's environmental distribution such as transmission routes (which includes studying the role of environmental factors like temperature, humidity, and the presence of other microorganisms), as well as impact on public health: The true extent of the public health threat posed by environmental A. baumannii strains remains unclear. Further research is needed to determine the potential for transmission from the environment to humans and the risk of acquiring infections with these antibiotic-resistant bacteria.
Minor points.
Figure 1; Discart > Discard
Line 152; host cell colonization?
Line 193-195; confusing sentence.
Line 215; delete one of the ‘community-acquired’
Line 285-287; confusing sentence.
AU: Thanks for your observation. All the typos were corrected, and the grammar and style were revised by experts from language editing services.
Round 2
Reviewer 3 Report
Comments and Suggestions for Authors
Accept in present form